# Analytical solution for nonadiabatic quantum annealing to arbitrary Ising spin Hamiltonian

Bin Yan [1,2] & Nikolai A. Sinitsyn [2✉]

Ising spin Hamiltonians are often used to encode a computational problem in their ground states. Quantum Annealing (QA) computing searches for such a state by implementing a slow time-dependent evolution from an easy-to-prepare initial state to a low energy state of a target Ising Hamiltonian of quantum spins, $H_I$. Here, we point to the existence of an analytical solution for such a problem for an arbitrary $H_I$ beyond the adiabatic limit for QA. This solution provides insights into the accuracy of nonadiabatic computations. Our QA protocol in the pseudo-adiabatic regime leads to a monotonic power-law suppression of nonadiabatic excitations with time $T$ of QA, without any signature of a transition to a glass phase, which is usually characterized by a logarithmic energy relaxation. This behavior suggests that the energy relaxation can differ in classical and quantum spin glasses strongly, when it is assisted by external time-dependent fields. In specific cases of $H_I$, the solution also shows a considerable quantum speedup in computations.

[1] Center for Nonlinear Studies, Los Alamos National Laboratory, Los Alamos, NM 87545, USA. [2] Theoretical Division, Los Alamos National Laboratory, Los Alamos, NM 87545, USA. ✉email: nsinitsyn@lanl.gov

The ground state of a classical Ising spin Hamiltonian $H_I(\sigma^1, \ldots, \sigma^N)$, where $\sigma^k$ are binary variables, can be found after QA by mapping $\sigma^k$ to the $z$-projection Pauli operators $\sigma_z^k$ of quantum spins-1/2 (qubits). The Hamiltonian for QA is generally defined as[1–4]

$$H(t) = f(t)H_I + r(t)H_M, \tag{1}$$

where $f(t)$ is monotonically increasing with time from zero to a finite value and $r(t)$ is monotonically decreasing from a finite value to zero; $H_M$ is the initial "mixing" Hamiltonian whose ground state is easy to prepare, and

$$H_I = \sum_k a_k \sigma_z^k + \sum_{k \neq s} a_{ks} \sigma_z^k \sigma_z^s + \sum_{k \neq s \neq r} a_{ksr} \sigma_z^k \sigma_z^s \sigma_z^r + \ldots \,. \tag{2}$$

The number of different terms in (2) can be exponentially large as $H_I$ can have arbitrary $k$-local terms that couple $k$ spins directly with different coefficients $a_{\{k\}}$.

Allowing only binary couplings in (2), this already includes NP-complete problems[5–8], which means that many important QA problems that are usually formulated with a different from (2) target Hamiltonian, can be mapped to the model (1) with only a polynomial overhead. The integer number factorization and the Grover algorithm can be also formulated as QA problems with some $H_I$[9,10].

Today, accessible hardware for a large number, over 100, qubits uses only heuristic approaches to QA[11], for which the operator $H_M$ and the annealing schedule, $f(t)$ and $r(t)$, in (1) are not specifically tuned to the choice of $H_I$. The QA protocol is chosen then mainly for the simplicity of implementing it in practice. Still, $H_M$ must not commute with $H_I$, and have a large gap between the lowest eigenvalue and the rest of its spectrum. According to the adiabatic theorem, if the time-dependent parameters change sufficiently slowly, the system remains in the instantaneous ground state and thus transfers to the ground state of $H_I$ as $t \to \infty$. Measuring the qubit polarizations $\sigma_z^k$, $k = 1, \ldots, N$, we then obtain the desired configuration of Ising spins that minimize $H_I$.

In real heuristic QA experiments, time is restricted by the coherence time of qubits, so the adiabatic regime is practically never achievable. Given the widths $\Delta E_I$ of the energy band of $H_I$, it is possible to perform a pseudo-adiabatic evolution with $T \gg 1/\Delta E_I$, where $T$ is the achievable QA time. However, the gap between nearest levels of $H_I$ is generally $\delta \sim \Delta E_I/2^N$, i.e., exponentially smaller than $\Delta E_I$, during the QA. The ground state of the full Hamiltonian $H(t)$ with a complex $H_I$ then usually passes through avoided crossings with exponentially small gaps to other levels. Hence, the practical situation corresponds to the nonadiabatic regime.

Thus, the experimentally accessible QA computing is inspired by a phenomenological assumption that there are computational problems whose partial solutions, i.e., the low Ising spin energy states can be obtained during the nonadiabatic QA process faster than during classical computations. If this assumption is correct, the quantum coherent evolution can be used in combination with incoherent classical annealing for a longer time.

Whether this is true or not is hard to verify either numerically or analytically because we deal with driven and nonadiabatic many-body dynamics. We still do not have definite answers on how quickly the useful information is gained during nonadiabatic QA computations, and whether there can be quantum algorithms that outperform classical computations during the time that is accessible in practice.

## Results

**Solvable model**. To address these problems, first, let us show that the original model (1) can be rewritten in the form of a scattering problem that depends on a single time-dependent parameter $g(t)$. In the Schrödinger equation,

$$i\frac{d}{dt}\psi(t) = H(t)\psi(t), \tag{3}$$

we switch to a new time variable

$$s(t) = \int_0^t d\tau\, f(\tau).$$

Here, $f(\tau)$ is positive, so $s(t)$ is a single-valued function, which is growing monotonically with $t$. Moreover, since both $f(t)$ and $r(t)$ are changing monotonically with $t$, they are single-valued functions of $s$: $f(s) \equiv f(t(s))$ and $r(s) \equiv r(t(s))$. Using that

$$\frac{d}{dt} = f(t(s))\frac{d}{ds}$$

in (3), we find that (3) is equivalent to

$$i\frac{d}{ds}\psi(s) = \left(H_I + \frac{r(s)}{f(s)}H_M\right)\psi(s). \tag{4}$$

Since $f(s) \to 0$ as $s \to 0$, the initial conditions become

$$g(s) \equiv \frac{r(s)}{f(s)} \to \infty, \quad \text{as} \quad s \to 0,$$

and since $r(s)$ decays to zero as $s \to \infty$, so does the redefined coupling $g(s)$. Thus, the QA problem in (1) is equivalent to a model with the Hamiltonian

$$H(t) = H_I(\sigma_z^1, \ldots, \sigma_z^N) + g(t)H_M(\{\boldsymbol{\sigma}\}), \tag{5}$$

where $g(t)$ is decaying from an infinite value to zero.

Next, if the goal is to study the accuracy of computations, one needs the probabilities of nonadiabatic excitations that are produced during QA starting from the ground state. Here, we point to the fact that there is a fully solvable model that provides all excitation probabilities for evolution (5) with an arbitrary $H_I$. This model has $g(t)$ and $H_M$, which satisfy the basic requirements for a QA protocol. Namely,

$$g(t) = -\frac{g}{t}, \quad g > 0, \tag{6}$$

and $H_M$ is the projection operator onto the state with all spins pointing along $x$ axis:

$$H_M = |\psi_0\rangle\langle\psi_0|, \quad |\psi_0\rangle \equiv |\to \cdots \to\rangle. \tag{7}$$

This $H_M$ has been considered for QA problems previously in relation to the adiabatic Grover algorithm[10]. In Methods, we show that the model remains solvable even when the state $|\psi_0\rangle$ is chosen arbitrarily. This means that the model generally depends on $2^N$ different complex parameters that encode this state. However, having no information about $H_I$, a wise choice of $H_M$ would be to consider $|\psi_0\rangle$ that does not discriminate among possible eigenstates of $H_I$, which is achieved by initially polarizing all spins along the $x$ axis.

As $t \to 0_+$, the state $|\psi_0\rangle$ is the ground state of $H$ with energy

$$E_0 \approx -\frac{g}{t}.$$

Since all the other eigenvalues of $H_M$ are zero, $|E_0|$ is also the leading order energy gap to the rest of the spectrum of $H$ as $t \to 0$.

Let $|n\rangle$ be the state of an arbitrary configuration of all the spins with definite projections along the $z$ axis. For this state,

$$|\langle n|\psi_0\rangle|^2 = \frac{1}{\mathcal{N}}, \quad \forall n, \tag{8}$$

where

$$\mathcal{N} = 2^N \tag{9}$$

is the dimension of Hilbert space of $N$ spins-1/2's. Thus, the matrix form of $H_M$ in the computational basis has identical exponentially small but nonzero entries. Let us also introduce the Ising energies

$$\varepsilon_n \equiv \langle n|H_I|n\rangle, \quad 0 \le n < \mathcal{N}, \tag{10}$$

where we reserve $n = 0$ for the ground state of $H_I$, and assume that the state indices are chosen so that

$$\varepsilon_0 < \varepsilon_1 < \ldots \varepsilon_{\mathcal{N}-1}.$$

We postpone the case of $H_I$ with eigenvalue degeneracy to a later section. We will call $n$ in $\varepsilon_n$ the number of excitations, because this index tells how many basis states have smaller Ising energy than the given state.

Let $a_0(t), \ldots, a_{\mathcal{N}-1}(t)$ be the amplitudes of the basis states in the Schrödinger equation solution:

$$\left|\psi(t)\right\rangle = \sum_{n=0}^{\mathcal{N}-1} a_n(t)|n\rangle.$$

For our QA protocol, the Schrödinger equation is given by

$$i\dot{a}_n = \varepsilon_n a_n - \frac{g}{\mathcal{N}}v, \quad v = \frac{1}{t}\sum_{k=0}^{\mathcal{N}-1} a_k, \quad n = 0, \ldots, \mathcal{N}-1. \tag{11}$$

The solvability of equations (11) follows from the fact that, after the Laplace transform, the $\mathcal{N}$ coupled equations reduce to a single first-order ordinary differential equation in the Laplace transform of $v$, which can always be solved analytically (see Methods). This model is a special case of a model that was solved by one of us[12]. Algebraic properties of this model were also mentioned in refs. [13,14], but the relation of its solution to the QA problem has not been discussed before.

The analytical solution gives a simple formula for the probabilities of excitation numbers at the end of the evolution. If as $t \to 0_+$ the system is in the ground state, $|\psi_0\rangle$, the probability to produce $n$ excitations as $t \to \infty$ is given by

$$P_n = \frac{p^n(1-p)}{1-p^{\mathcal{N}}}, \quad p \equiv e^{-\frac{2\pi g}{\mathcal{N}}}. \tag{12}$$

Note that the final state probabilities do not depend on the particular expressions for the eigenstates $|n\rangle$, and in this sense tell nothing about the ground state of $H_I$. However, equation (12) gives complete information about the performance of the given QA protocol. For example, the probability to obtain the ground state is given by

$$P_0 = \frac{1-p}{1-p^{\mathcal{N}}}, \tag{13}$$

and the average number of excitations is

$$\langle n \rangle \equiv \sum_{n=1}^{\mathcal{N}-1} nP_n = \frac{p}{1-p} + \left(1 - \frac{1}{1-p^{\mathcal{N}}}\right)\mathcal{N}. \tag{14}$$

These expressions simplify for a large number of interacting qubits $N \gg 1$, for which $\mathcal{N}$ is exponentially large, and we can disregard $p^{\mathcal{N}}$ in comparison to $p$. For $g \gg 1$ we find $p^{\mathcal{N}} \ll 1$, and $P_n$ follows the geometric distribution, with

$$P_0 = 1 - p, \quad \langle n \rangle = \frac{p}{1-p}. \tag{15}$$

To provide an intuition about the properties of the distribution (12), we also note that if the energy dispersion of $H_I$ were linear, i.e., if $\varepsilon_n = n\delta$, then the distribution (12) would be the Gibbs distribution

$$P_n = \frac{e^{-\varepsilon_n/(k_BT)}}{Z}, \quad \varepsilon_n = n\delta,$$

where $1/Z$ is a normalization factor and

$$k_BT = \mathcal{N}\delta/(2\pi g) \sim \Delta E_I/g. \tag{16}$$

As the dimensionless parameter $g$ is growing, the effective temperature (16) of the final excitation distribution is decreasing.

**Characteristic annealing times.** The currently studied QA systems use a slowly changing transverse magnetic field with

$$H_M^0 \equiv -\sum_{k=1}^{N} \sigma_x^k, \tag{17}$$

where $\sigma_x^k$ are Pauli $x$-operators acting in space of individual spins. In later sections, we will argue that the model with schedule $g(t)$ in (6) and $H_M$ from (7) is, for a certain large subclass of $H_I$, optimal. Therefore, its solution can be used to learn about the entire strategy of using nonadiabatic QA for finding low-energy states. To show this, we must first introduce a method to compare the performance of different QA protocols with $g(t) \sim 1/t^{\alpha}$ and different $H_M$, but the same $H_I$ and the computation time $T$.

There is an additional time scale that characterizes the speed of QA. The operator $g(t)H_M$ has a bounded spectrum. Due to the exponentially large Hilbert space, this spectrum must have a high-density region at some distance $\Delta E_M$ from the ground state of $g(t)$ $H_M$. The Ising part $H_I$ also has a characteristic energy scale $\Delta E_I$, that is, the bandwidth of its spectrum (Fig. 1, left panel). Since $H_I$ and $g(t)H_M$ do not commute, the resonant nonadiabatic transitions between the ground level of $g(t)H_M$ and the dense region of its spectrum become most probable near the time $\tau_a$, when the operators $H_I$ and $g(\tau_a)H_M$ become comparable (Fig. 1, left and middle panels), i.e.,

$$\Delta E_M(\tau_a) = \Delta E_I. \tag{18}$$

For example, for our solvable model (see Methods)

$$\tau_a = g\tau_I,$$

where

$$\tau_I = 1/\Delta E_I$$

is the characteristic time of dephasing that can be induced by the Ising part $H_I$. We will call $\tau_a$ the annealing time, in contrast to the total evolution time $T$ that we will call computation time.

Any QA protocol must pass through the moment (18). Hence, $\tau_a$ can always be defined consistently. We will say that two different protocols with power-law decays of $g(t)$ and the same $H_I$ and $T$, have the same speed of QA if they also have the same $\tau_a$. The practically interesting values of $\tau_a$ are restricted to the range

$$\tau_I < \tau_a < T. \tag{19}$$

The first inequality in (19) follows from the fact that the case of $\tau_a < \tau_I$ corresponds to a strongly nonadiabatic regime, for which the gap in the spectrum of $g(t)H_M$ closes faster than the characteristic interaction rates of $H_I$. We will say that one of the compared protocols is better if it produces fewer excitations, $\langle n \rangle$, when $T/\tau_a = \text{const} \gg 1$ and the same characteristic times, $\tau_I$ and $\tau_a$, are set for the different protocols.

If a protocol is optimal, i.e., outperforms all other protocols at some imposed conditions on the QA schedule and for a certain class of $H_I$, it must remain optimal after time-rescaling, $t \to \lambda t$, in the Schrödinger equation, because the latter merely means the change of time-counting procedure. It has been recently proved[15] that if such a protocol exists, it must correspond to a power-law decay of the coupling: $g(t) \sim t^a$. We will use this result because it strongly restricts the class of the schedules that should be tested in order to prove the optimality. Here we also note that the solvable protocol has $g(t) \sim 1/t$, which means that it may be optimal for some classes of $H_I$, which we will identify later.

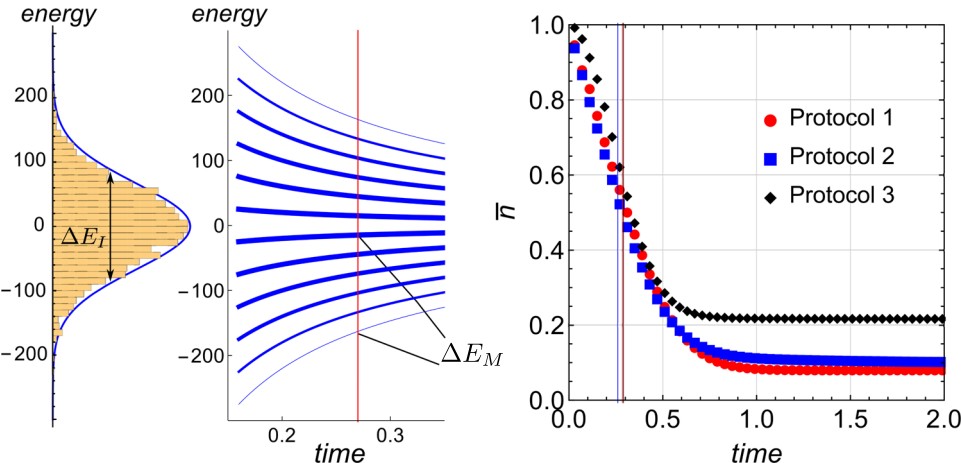

**Fig. 1 Characteristic annealing time.** Left: histogram of the spectrum of the Hamiltonian $H_I$(2) with Gaussian distribution of all coefficients $a_{\{k\}}$. Middle: time evolution of the spectrum of the transverse field Hamiltonian $H_M^0$(17) with quench schedule $g(t) = g/t$, where $g = 4$. (Thicker curves correspond to the higher density of states). The vertical line marks the annealing time $\tau_a$(18), for which the gap between the ground state and the highest density point in the spectrum equals the bandwidth $\Delta E_I$ (full width at half maximum) of the spectrum of $H_I$. Right: time evolution of the (normalized to 1) excitations $\bar{n} \equiv \langle n \rangle / ((\mathcal{N} - 1)/2)$ for different QA protocols at $g = 4$ of $N = 12$ spins. These protocols are tuned to have the same annealing time $\tau_a$ as summarized in Table 1. The colored vertical lines mark the corresponding times at which the excitations reach the halfway into their saturation, which verifies almost the same effective annealing rate for the protocols with the same $\tau_a$.

**Computational convergence rates.** The analytical solution says that the probability to find the ground state configuration is growing linearly with $\tau_a$, however, starting from an exponentially small value. Thus, if we assume that $g = \tau_a/\tau_I \gg 1$, then

$$P_0 = 1 - p \approx \frac{2\pi g}{\mathcal{N}} = \frac{2\pi}{\mathcal{N}} \frac{\tau_a}{\tau_I}.$$

Hence, in order to make $P_0 \sim 1$, we need the QA time

$$\tau_a \sim \tau_I 2^N. \tag{20}$$

The theory of simulated QA has previously produced various bounds on the rate of change of the coupling[16–18]. The simulated QA is a Monte-Carlo algorithm, which performance dependence on $N$ and $T$ can be different from the performance of the physical QA but both algorithms are interesting to compare. According to ref. [17], to guarantee the convergence of the simulated QA for binary couplings in the Ising Hamiltonian, as $t \to \infty$, to $O(1)$ ground state probability, the field should change as

$$g(t) \geq (\xi t)^{-1/(2N-1)}, \tag{21}$$

where $\xi$ is exponentially small for large $N$. Our solution agrees with this estimate. It shows the convergence of QA computing to the ground state in the adiabatic limit, during a finite non-polynomial in $N$ annealing time (20). However, for a fair comparison, the result in ref. [17] must be extended to the limit of maximal complexity of (2). At least the fact that the number of terms in $H_I$ can be exponentially large adds an extra-large overhead on the Monte-Carlo algorithms, such as the simulated QA, because the time to calculate just one eigenvalue becomes, itself, exponentially large. In the worst-case then, the calculation time should grow as $\sim \mathcal{N}^2$. In contrast, programming such a complex $H_I$ for QA means setting $O(\mathcal{N})$ different couplings only once. This takes only $O(\mathcal{N})$ amount of time and therefore this preparation step for QA can change only the exponential prefactor but not the exponential scaling in (20).

The result (20) also shows that the generally exponentially hard computational problem requires exponentially large calculation time for a precise solution. Hence, computational difficulties reemerge in some form in different computational approaches. For specific problems, this annealing time can be generally obtained by the gap analysis and fine-tuning of the protocol for a specific $H_I$. For example, if the minimal gap over the ground state scales as $\sim 1/\sqrt{\mathcal{N}}$, this imposes the same constraint for the annealing time $\tau_a \sim \mathcal{N}$. However, we stress that the gap analysis for complex $H_I$ can be very challenging, and a proper choice of the annealing protocol, $g(t)$ and $H_M$, requires individual tuning[19,20]. In contrast, our analytic solution applies to all $H_I$ with a fixed simple form of the annealing protocol.

The time estimate (20) can be compared to the one for a classical search algorithm that would identify the ground state of the diagonal matrix $H_I$. If the entries of $H_I$ are random, there is no other way but to compare all eigenvalues, which requires $\mathcal{N}$ computational steps. Using this analogy, equation (20) suggests that $\tau_I$ can be considered as an analog of the single computation time step and $\tau_a$ is the analog of the full computation time in the classical search algorithms.

**Scaling for the average excitation number.** The modern attempts to develop QA hardware are largely based on a heuristic assumption that at moderate QA rates we can obtain a considerable reduction in computational error rate even when the true ground state cannot be found. The needed intuition for this regime can be gained from physics using the similarity of the complex Ising Hamiltonians with spin-glass systems that correspond to randomly chosen couplings between spins[21]. The glass phase appears at low temperatures and corresponds to logarithmically slow relaxation of standard measurable characteristics[22]. Indeed, classical annealing simulations of spin glasses generally show a logarithmic residual energy dependence on time $T$ of the temperature decay from a finite value to zero[16,23]:

$$\varepsilon_{res}^{cl} \sim 1/\log^\beta T, \tag{22}$$

where $\beta = O(1)$ is a constant. The transition to the glass phase is also expected for QA but the scaling of the residual energy with QA time is not clear. On one hand, quantum tunneling is more efficient than thermal fluctuations when overcoming spikes of a potential barrier. On the other hand, such barrier spikes can be bypassed in the multidimensional phase space of many qubits, whereas stochastic fluctuations are more efficient for transiting over shallow but broad potential barriers. Moreover, disordered

quantum systems show purely quantum effects, such as many-body localization, that resist the propagation of information inside a system. An example of this behavior is found in gamma-magnets[24]—the models of arbitrarily many interacting spins that resist flipping even a single spin in response to arbitrarily strong and fast magnetic fields. Thus, there are arguments both in favor and against QA in comparison with classical annealing performance.

Early numerical studies found that QA leads to an inverse power of the logarithmic decay (22) as well, where $T$ is the time of the QA protocol, but with a larger power $\beta$, and hence outperforms classical annealing[25,26]. However, later studies[27] claimed that this behavior might be a numerical artifact caused by time discretization, and the improvement of QA reduces only to a small finite offset in the time-continuum limit. If the system passes into a glassy phase, there are analytical arguments showing that QA has no advantage over classical annealing at all[28]. In any case, if slow energy relaxation (22) describes QA of spin glasses in the pseudo-adiabatic regime generally, the heuristic QA method looks impractical for computations, apart from niche applications that avoid the spin-glass behavior.

Returning to our solvable model, QA superiority in the nonadiabatic regime would correspond to a fast suppression of the average number of excitations, for $\mathcal{N} \gg 1$, which is given by

$$\langle n \rangle = \frac{p}{1-p} \approx \frac{1}{P_0} = \mathcal{N} \frac{\tau_I}{2\pi\tau_a}. \tag{23}$$

As expected, $\langle n \rangle$ decreases with the growing annealing time $\tau_a$ but nonexponentially and starting from an exponentially large initial value.

Let us now discuss the fact that, formally, the computation time $T$ in the solvable model is infinite but in practice, it has to be finite. Let us set $T$ to be proportional to $\tau_a$. The same scaling then would be found for the dependence of $\langle n \rangle$ on $T$ if the deviation of the QA result at finite $T$ from the exact solution is suppressed by a small parameter $\tau_a/T$. Numerically, we always found that $\langle n \rangle$ saturates for $T > \tau_a$ close to the $T \to \infty$ value, up to corrections of some order of $\tau_a/T$ (Fig. 1, right panel).

The following analytical arguments show that, indeed, a sudden termination of the protocol at finite $T \gg \tau_a$ produces a negligible difference from our analytical prediction. Using the Landau–Zener formula, the nonadiabatic transitions may not be suppressed during $t > T$ for the states within the energy difference $\delta\varepsilon^2 \sim |d\Delta E_M/dt| \leq g/T^2$. For spin glasses with a smooth density of states, the introduced deviations from $\langle n \rangle$ are suppressed, at least, by a factor $O(\tau_I/T)$, which has the same dependence on $T$ as the $\langle n \rangle$ dependence on $\tau_a$ but the factor $1/T$ is much smaller. For example, if we set $\tau_a/T \sim 0.01$, then the deviations from the analytical prediction for $\langle n \rangle$ should not exceed $\sim 1\%$. Thus, we find the scaling

$$\langle n \rangle \sim 1/T, \tag{24}$$

assuming that $\tau_a/T = \text{const} \ll 1$.

Equation (24) is the main result of our article. We showed analytically that QA with the solvable protocol does not lead to a logarithmically slow relaxation for arbitrarily complex $H_I$. In fact, the exact solution does not show any sharp changes in the relaxation curve, which are expected for the transition to a glass phase.

We now analyze the behavior of the residual energy

$$\varepsilon_{res} = \langle H_I \rangle_{t \to \infty} - \varepsilon_0. \tag{25}$$

For spin glasses with random $H_I$, the middle of the density of states is smooth and broad, and can be well described with a constant density, i.e., $E_n = \delta n$, where $\delta = \Delta E_I / \mathcal{N}$ is the characteristic distance between nearest energy levels. In this case,

for a broad range of annealing times, $\langle n \rangle$ and the average energy after QA are linearly related: $\varepsilon_{res} \sim \langle n \rangle \delta$. Then, equation (23) means a surprising fact that the energy relaxation as a function of the annealing time follows a power law:

$$\varepsilon_{res} \sim \Delta E_I(\tau_I/\tau_a), \tag{26}$$

rather than a logarithmic relaxation with growing $\tau_a$, which is found in the classical annealing of spin glasses.

In interacting-spin systems, the density of state typically follows a Gaussian form[29], whose tail near the ground state can be distorted, e.g., to an exponential shape. Hence, for truly slow QA, deviations from (26) are expected because the residual energy becomes sensitive to the exact form of the density of states near the ground level. However, any power-law energy dispersion near the ground level, $\varepsilon_n \sim n^\alpha$, leads to a power law $\varepsilon_{res} \sim 1/\tau_a^\alpha$ rather than logarithmic residual energy dependence on $1/\tau_a$ after averaging over the distribution (12). This allows us to analyze the residual energy scaling with various forms of low-energy spectral density. In Methods, we show that the power-law relaxation for the residual energy is typically expected, including for the Gaussian and exponential spectral densities.

Numerically, we could not find a spectrum that would produce a clearly logarithmic residual energy relaxation for the solvable excitation distribution. We attribute this to the fact that the inverse power law for the average excitation (24) is a sufficiently strong constraint to lead to a power-law relaxation for a broad type of energy spectra. We leave the question open: whether this behavior is a consequence of the non-local nature of the mixing Hamiltonian (7).

Below, we discuss other properties of the solvable protocol, which should be of interest for the heuristic QA hardware developments.

**Degenerate ground state**. The exponentially large QA time is needed for the solvable protocol to obtain the ground state only if this state is nondegenerate. We consider now the case with the ground state degeneracy: $\varepsilon_1 = \ldots = \varepsilon_{M-1} = \varepsilon_0$. Summing the first $M$ equations in (11), we then find that the superposition

$$|+\rangle = \frac{1}{\sqrt{M}} \sum_{m=0}^{M-1} |m\rangle$$

is coupled to any $|n\rangle$, where $n \geq M$, with a larger coupling $g\sqrt{M}$. All other orthogonal superpositions of the Ising ground states then decouple and have zero probability to be at the end of the evolution.

The solvable model in Appendix B of ref. [12] (see also Methods) is applied even when all $\mathcal{N}$ states are coupled to each other with different independent $\mathcal{N}$ parameters. Thus, the modification of the effective coupling to state $|+\rangle$ is still described by the exact solution in ref. [12], which leads to the probability of the final state $|+\rangle$:

$$P_+ = (1 - p^M)/(1 - p^{\mathcal{N}}), \quad p = e^{-\frac{2\pi g}{\mathcal{N}}}, \tag{27}$$

whereas the probabilities of the energy excitations do not change. This gives us an estimate for the time to prepare the state $|+\rangle$ with probability $P_+ \sim 1$:

$$\tau_a \sim \frac{\mathcal{N}\tau_I}{M}.$$

If $M$ is large, e.g.,

$$M \sim \mathcal{N}/\log_e^a \mathcal{N}, \tag{28}$$

this leads to an exponential speedup for extracting non-local information that can be obtained from measurements on the prepared superposition $|+\rangle$.

For example, suppose that all excitation energies of $H_I$ are random positive and $\varepsilon_0 = \ldots = \varepsilon_{M-1} = 0$ appear periodically, so that, when sorted in the known standard computational basis, they correspond to the eigenstates $|x_0 + rT\rangle$, where $x_0$ and $T$ are integers, such that $x_0 < T \sim \log_e^a \mathcal{N}$; $r = 0, 1, 2, \ldots$, and $\mathcal{N}/T$ is also an integer. This corresponds to $M \sim \mathcal{N}/\log_e^a \mathcal{N}$, so during the QA time of an order

$$\tau_a \sim \tau_I \log_e^a \mathcal{N}$$

the solvable protocol prepares a state of the qubits as a symmetric superposition:

$$|+\rangle = \frac{1}{\sqrt{M}} \sum_r |x_0 + rT\rangle. \tag{29}$$

The Quantum Fourier Transform then can be used to change this state into a superposition of the states $|k\rangle$, where $k$ is the integer multiple of $\mathcal{N}/T$. Finding only two different $k$, one can then find their greatest common divisor by classical means, and thus determine the period $T$ faster than by classical means.

The possibility to solve the period finding problem on a quantum computer is an essential ingredient in many quantum algorithms, such as Shor's factorization algorithm. An important step in such algorithms is to find a symmetric superposition of equal energy eigenstates of a quantum function that has a high degeneracy of eigenstates in the entire phase space. Such a function can be usually encoded in the target Hamiltonian $H_I$ and thus one of its eigenstates can be found using QA. However, it is clear from our solution why such algorithms are hard to implement with other heuristic protocols, such as with the transverse field (17). This field couples different Ising ground states with the higher Ising energy states differently. Hence, even if we assume that the ground state can be prepared quickly, it will appear generally in a nonsymmetric superposition

$$|\psi_G\rangle = \sum_r C_r |x_0 + rT\rangle,$$

where the coefficients $C_r$ have not only different absolute values but also different phases which depend on all parameters of $H_I$. Hence, further manipulations, such as making the Quantum Fourier Transform, may not provide the desired effect on this state, which is needed to complete the algorithm.

**Effectiveness of the solvable protocol in the limit of maximal complexity of $H_I$.** The annealing protocol in our solvable model is unbiased in the sense that the amplitudes $a_n(t)$ (11) do not depend on the specific structure of the basis states. This is not the case for the protocol with a transverse field[30], which couples directly only to the basis states whose net spin polarization differs by ±1. Our protocol is also unbiased in the sense that degenerate ground state configurations as a symmetric superposition couple to the other states equally, which results in equal probabilities to find such ground states of $H_I$.

Moreover, the statistical learning theory[31] says that direct approaches, which avoid the gain of irrelevant information, should be favorable for learning algorithms. This is partly addressed by our finding that the final state probabilities obtained by solving equation (11) are independent of the precise values of $\varepsilon_k$, i.e., the transition probability to any state $|n\rangle$ depends only on how many other states have smaller Ising energies. For example, the probability to find the ground state does not depend on the choice of $H_I$ at all. This independence of the scattering probabilities of certain basic parameters is shared by all integrable models with time-dependent Hamiltonians[13] but is not expected otherwise. Hence, it must be unique for $g(t) \sim 1/t$ annealing protocol because other $g(t)$ is not among the known solvable models with arbitrary $H_I$. This property means that our solvable protocol does not produce irrelevant information about specific values of $\varepsilon_k$, as needed because only the ordering of these eigenvalues matters for finding good approximations to the ground energy.

Such properties altogether are unique among the possible QA protocols, which suggests that the solvable protocol, for some types of problems, could be favorable. Owing to the universality of the analytical solution, if true, this should be true for the most complex form of $H_I$. Thus, let $H_I$ be the sum of all possible terms in (2) with independent random coefficients $a_{\{k\}}$. Such a high-complexity limit reduces the problem of identifying the minimal value from an unsorted array of independent random energies $\varepsilon_n$ that are sampled from some distribution. For instance, for Gaussian random coupling coefficients, $\varepsilon_n$ forms a Gaussian distribution as well (Fig. 1, left panel). Such a construction of $H_I$ does not favor any particular ground state spin configuration and even any systematic correlations between the excited states. Hence, it is expected that the low-energy states are estimated faster with a maximally unbiased QA protocol, which is our solvable protocol.

To test this hypothesis, we employ the result in[15] that allows us only to compare the performance of the solvable protocol with a family of the protocols with a power-law decay of the coupling, $g(t) \sim 1/t^\alpha$, and identical for each protocol fully random $H_I$, as well as $\tau_a$ and $T/\tau_a$. First, we note that the protocols with $\alpha < 1$ produce definitely worse than $\langle n \rangle \sim 1/\tau_a$ scaling for the excitations if we set $T/\tau_a = \text{const}$. This follows from the fact that even in the adiabatic approximation the term $H_M/t^\alpha$ mixes any Ising eigenstate with other states within the window of energy $\varepsilon \sim 1/t^\alpha$. Hence, sudden termination of such protocols at a finite time $T$ cannot resolve the states within the energy window that scales as $1/T^\alpha$, which decays slower than $1/T$.

For $\alpha \geq 1$, we resort to the numerical investigation. Figure 2 compares numerically calculated final $\langle n \rangle$ for different protocols at $N = 12$ and the Hamiltonian (2) with randomly chosen all possible couplings. For large $g$, which we define for all protocols as $g \equiv \tau_a/\tau_I$, the excitation number decays as a power law. For any $g$ and $N$, our analytically solvable model (Protocol 1) always outperforms the other protocols, although all of them show scaling similar to $1/g$ for large $g$. In numerous other tests (not shown), we found that all non-power-law schedules, e.g., with $g(t)$ decaying exponentially, had a much worse performance for the same values of $\tau_I$, $\tau_a$, and $T$, in agreement with[15]. Figure 3 also shows the data that we used to extrapolate the results to larger $N$. For such interpolations, we always found that the solvable protocol produced smaller residual energy for the fully random Hamiltonian $H_I$. Hence, as far as we could test numerically and extrapolate our results, the solvable protocol was, indeed, optimal for our comparison criteria and the most complex form of $H_I$.

An alternative argument for the optimality of the solvable protocol for fully random $H_I$s follows from the estimate (23), which says that the performance of this protocol is actually the same as in the classical Monte-Carlo search. Indeed, a random search for the lowest eigenvalue has probability $n_{\max}/\mathcal{N}$ per step to pick up an eigenvalue from the first $n_{\max}$ excitations. Hence it takes time $\tau \sim \mathcal{N}\tau_{\text{step}}/n_{\max}$ to find an eigenvalue with $0 \leq n \leq n_{\max}$, where $\tau_{\text{step}}$ is the time of one eigenvalue of $H_I$ computation and its comparison to a previously found lowest value. This is precisely the estimate of equation (23), in which we identify $\tau_a$ with $\tau$, $\tau_I$ with $\tau_{\text{step}}$ and $\langle n \rangle$ with $n_{\max}$. Since our QA protocol has the same convergence rate as the classical Monte-Carlo search of the completely unsorted array, any improvement over its performance on $H_I$ with all random entries, either for the full or the partial search, would mean the quantum supremacy that does not rely on hints such as the oracle in the Grover algorithm, which is believed to be impossible.

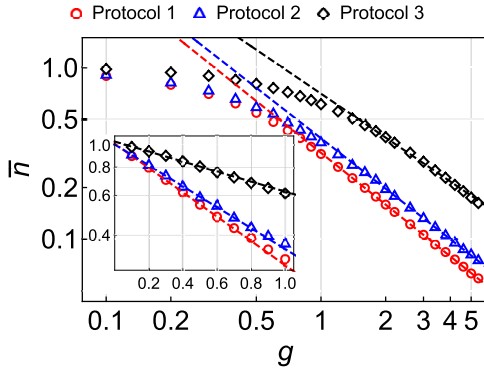

**Fig. 2 Scaling of the excitation number.** Numerically found final normalized excitation number $\bar{n} \equiv \langle n \rangle / ((\mathcal{N} - 1)/2)$ at various $g = \tau_a/\tau_l$ for $N = 12$ spins for three different protocols listed in Table 1. The Hamiltonian $H_I$ takes the form (2) with the coupling coefficients independently drawn from the standard normal distribution. The main figure and inset show the adiabatic (large $g$) and nonadiabatic (small $g$) regimes in log–log and semi-log scales, respectively. The solvable protocol (red points) always outperforms the other protocols for the same $g$.

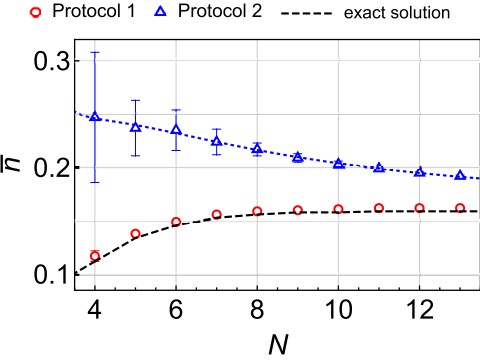

**Fig. 3 Asymptotic excitation numbers.** Numerically found final normalized excitation numbers $\bar{n} \equiv \langle n \rangle / ((\mathcal{N} - 1)/2)$ for various numbers of spins at $g = 2$ and Ising Hamiltonian (2) with the coupling coefficients independently drawn from the standard normal distribution. Blue dots correspond to the transverse field protocol, and red dots are numerical confirmation of the analytical prediction for the solvable model, which is marked by a dashed black curve. Error bars show the standard deviations of the data points obtained from averaging over 25 realizations of the random Hamiltonian. A Blue dotted curve is the best fit to $\sum_{k=0}^{4} c_k / N^k$, which predicts that the exact solution has better performance for any $N$.

Thus, our protocol gives an explicit example of heuristic QA computations leading to the same performance as for one of the known classical algorithms. This includes all possible $H_I$ with nondegenerate spectra, and all possible time restrictions. As our QA protocol, the unbiased random search Monte-Carlo is the preferable choice for searching through a completely random array but then by classical means. This raises a question of whether many other heuristic approaches, such as using the practically most accessible QA protocols without correlating them with the desired task, or post-processing the final state as in the case of the ground state degeneracy, have also the same performance for all possible tasks as certain classical algorithms.

**Avoiding the bound.** The limit of fully random $H_I$ represents the largest class of all possible computational problems (5). Classical optimization algorithms usually trade between good and bad performance in different applications, which is known as the "no-free-lunch" property. Although similar results are not known for QA, it is expected that the effectiveness of the solvable protocol for the big class of the most complex problems generally means that there are protocols that outperform it on simpler problems with more structured $H_I$. Below, let us show several examples in support of this hypothesis.

A well-known example of a problem with a structured $H_I$ is the one that is solvable by the Grover algorithm. It prepares the ground state of an operator $H_I$ that has all but one zero eigenvalues, whereas the ground state energy is $-1$. Let $\eta_k = \pm 1$, where the sign depends on whether this ground state has the $k$-th spin, respectively, up or down. Then, $H_I$ for Grover's problem can be written as

$$H_I^G = -\prod_{k=1}^{N} \frac{(1 + \eta_k \sigma_z^k)}{2}. \tag{30}$$

In comparison with the most complex version of (2), this Hamiltonian is much simpler. It depends only on $N$ sign parameters, and it has considerable symmetry: changes in these parameters do not affect the spectrum of $H_I^G$. It is, indeed, known that the ground state of $H_I^G$ can be found by adiabatic QA during the time that scales only as $\mathcal{N}^{1/2}$[10]. Achieving this adiabatically requires a very fine-tuned choice of the schedule $g(t)$. However, if our solvable protocol is not optimal for the structured problems there must be protocols that achieve better estimates for the ground state for Grover's problem also beyond the adiabatic regime, and such protocols may not need to be very complex.

Let us show that this expectation is true. Consider the QA Hamiltonian

$$H(t) = \varepsilon H_I^G - g(t) H_M, \tag{31}$$

where $H_M$ is given by (7). Due to the degeneracy of eigenvalues of $H_I^G$, the evolution equation (11) reduces to two coupled differential equations for the amplitude $a_0$ of the ground state and the normalized sum of the other amplitudes:

$$a_+ \equiv \frac{1}{\sqrt{\mathcal{N} - 1}} \sum_{k=1}^{\mathcal{N}-1} a_k.$$

Namely,

$$
\begin{aligned}
i\dot{a}_0 &= -\left(\frac{g(t)}{\mathcal{N}} + \varepsilon\right) a_0 - \frac{\sqrt{\mathcal{N} - 1}\, g(t)}{\mathcal{N}} a_+, \\
i\dot{a}_+ &= -\frac{(\mathcal{N} - 1) g(t)}{\mathcal{N}} a_+ - \frac{\sqrt{\mathcal{N} - 1}\, g(t)}{\mathcal{N}} a_0.
\end{aligned} \tag{32}
$$

The initial conditions, as $t \to 0_+$, correspond to $a_0 = 1/\sqrt{\mathcal{N}} \approx 0$ and, hence, $a_+ \approx 1$. The protocol that makes $P_0 \equiv |a_0|^2 \sim 1$ is obtained by immediately setting the schedule to a constant value

$$g(t) = \varepsilon \mathcal{N} / (\mathcal{N} - 2) \approx \varepsilon, \tag{33}$$

and then letting the system evolve under such conditions during time

$$T = \frac{\pi \sqrt{\mathcal{N}}}{2\varepsilon}. \tag{34}$$

One can verify that this makes $P_0 \approx 1$ by noting that equations (32) with condition (33) are equivalent to the evolution equations for a spin 1/2 in a transverse magnetic field, which rotates this spin. Condition (33) is needed to remove the component of this field that points along the spin axis. Time $T$ corresponds to a rotation angle that switches between orthogonal states of this spin.

Unlike the time of the solvable protocol with $g(t) = -g/t$, which scales as $T \sim \mathcal{N}$, the time in (34) scales as $\sim \sqrt{\mathcal{N}}$, which is expected for Grover's computational problem.

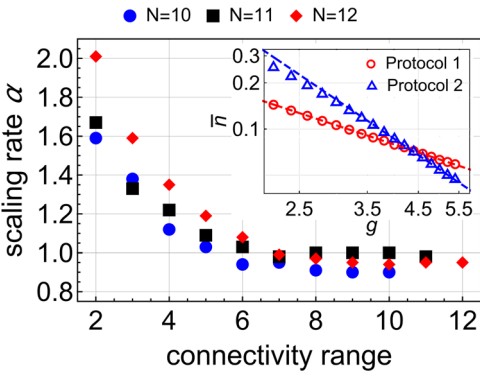

**Fig. 4 Annealing for systems with the finite connectivity range.** The scaling rate, $\alpha$, was obtained by fitting numerically obtained excitation numbers with $\langle n \rangle \sim 1/\tau_a^\alpha$, for various interaction ranges of the Hamiltonian $H_I$ and the transverse field protocol (Protocol 2 in Table 1). The inset shows a comparison of the typical power-law decay for such fits of the data for protocols 1 and 2 with $N = 12$ spins under a range-2 (only binary spin-spin couplings) Hamiltonian.

This efficient protocol to solve Grover's problem is fine-tuned for $H_I^G$ and cannot show good performance on other tasks. Identifying such algorithms for heuristic computations requires additional optimization steps, e.g., using the methods of machine learning[32], which would correlate the annealing protocol to a given structured $H_I$. Such methods, however, become inefficient in the limit of maximal complexity with fully random $H_I$ because of the emergence of the barren plateau[33].

Another example corresponds to the systems with small connectivity between qubits in $H_I$. It is expected then that a QA protocol that emphasizes interactions without many direct spin flips can achieve a better performance, such as the protocol induced by the decaying transverse field.

To test this, we performed simulations for $H_I$ with limited connectivity ranges, i.e., a range-$k$ Hamiltonian is of the form (2) but only contains terms with at most $k$ simultaneously coupled spins. This allows the control of the problem complexity by tuning the connectivity range. Our numerical simulations (Fig. 4) show that, for finite size systems of up to 12 spins and the transverse field (17), the final excitation numbers always scale as a power law of $g$, i.e., $\langle n \rangle \sim 1/g^\alpha \sim 1/\tau_a^\alpha$, and $\alpha$ increases with the decrease of the connectivity range of $H_I$. At $k = 2$, which is known as the Sherrington–Kirkpatrick model[34], $\alpha$ reaches the value 2.

Figure 4 demonstrates the convergence of the performance to the universality domain of the solvable protocol with increasing complexity. In agreement with our expectations, as far as we could see numerically, the protocol with the decaying transverse field produced better performance on the structured problems than the solvable protocol, in agreement with the "no-free-lunch" property.

Let us now return to the question of whether QA computations in the nonadiabatic regime can provide a better performance, in terms of scaling with the number of qubits, than the adiabatic quantum computations for the same problem. Our solvable protocol, as well as the nonadiabatic Grover protocol, do not show this feature, as their performances scale equally with the adiabatic QA. Generally, this may not be true.

Here, we note that there is one more solvable model of QA that can be used to explore the scaling of $\tau_a(N)$ for a specific simple $H_I$: Consider

$$H_I^\varepsilon = \sum_{k=1}^N \varepsilon_k \sigma_z^k, \tag{35}$$

that is subject to a non-local constraint $\sum_{k=1}^N \sigma_z^k = 0$. Let us assume that $|\varepsilon_k|$ are of the order $\varepsilon$. The ground state of $H_I^\varepsilon$ has $N/2$ spins pointing up. They correspond to the smaller half of $\varepsilon_k$ values. The other $N/2$ spins point down. Here, $H_I$ is parametrized by only $N$ numbers $\varepsilon_k$. Naturally, a wise algorithm should not look through all $2^N$ eigenvalues of $H_I$ but rather learn those parameters.

Due to the constraint, the ground state of (35) has zero total qubit polarization. To find this state, one can use the protocol with $H_M$ that also has the ground state with zero initial total spin[35]:

$$H(t) = \sum_{k=1}^N \varepsilon_k \sigma_z^k - \frac{g}{t} \sum_{i \neq j=1}^N \hat{\sigma}_i^+ \sigma_j^-. \tag{36}$$

As $t \to 0$, the ground state energy of $H(t)$ is separated from the dense region of $g(t)H_M$ near-zero energy by $\Delta E_M \sim \frac{gN(N-1)}{2t}$, and the $H_I^\varepsilon$ bandwidth scales linearly with $N$: $\Delta E_I \sim \varepsilon N$. The exact solution of this model was found in ref. [35]. It says that the ground state is determined if $g \approx 1$.

Using our definition of the annealing time, we can now compare the performance of such QA computations with the performance of classical algorithms for the same problem. We find for the model (36) that $g \approx 1$ corresponds to

$$\tau_a \sim N/\varepsilon.$$

The same solution in ref. [35] also shows that if we need only a partial search by allowing a fraction $\alpha \ll 1$ of mistakes, i.e., allowing $\alpha N$ spins pointing in a wrong direction, then it is sufficient to choose $g \sim 1/(N\alpha)$, i.e., the computation time reduces by a factor $\sim 1/(N\alpha)$, so in our notation

$$\tau_a \sim O(1/\alpha).$$

Classically, finding the smaller half of $N/2$ of $\varepsilon_k$ values takes $\sim N$ steps. The partial QA solution thus has a better $N$-scaling than both the best available classical algorithm and the complete solution in the adiabatic limit. This example supports the speculations that a hybrid approach that involves a moderately fast QA step combined with a subsequent classical relaxation may improve the search for the true ground state.

**Estimates for the physical time of computation.** The tests of QA hardware[36–41] on specific problems gave contradictory results. There are claims for superior performance of QA in some instances[42], but achieving scalable quantum supremacy[11] using QA is still far from conclusive.

Let us estimate the performance of our solvable protocol at the current level of technology. The coupling energy of a single qubit to the rest of the quantum processor is physically restricted to some value $\epsilon_{\max}$. For example, for a superconducting qubit, a coupling larger than the superconducting gap may produce unwanted excitations outside the qubit phase space. The bandwidth for $H_I$ is then restricted by $\Delta E_I < \epsilon_{\max} N$. Hence, $\tau_I$ for $N$ qubits is restricted by

$$\tau_I > 1/(\epsilon_{\max} N).$$

If we assume $\epsilon_{\max} = 10$GHz as the upper bound for a superconducting qubit, then to find the ground state of only 20 qubits, from (19), we need at least time $\tau_a \sim 0.1$ μs, which is the typical upper bound on coherence time of such qubits. The required computation time $\tau_a$ is growing exponentially with extra qubits, so chances to solve an optimization problem for >20 qubits with the modern level of quantum technology are quickly vanishing.

One practical advantage of the solvable protocol that may justify the efforts to implement it in hardware may follow the

complexity to retrieve the $H_I$ eigenvalues. Namely, when the sorting problem is encoded in the Hamiltonian of spin projection operators the direct classical algorithm requires the additional computation of eigenvalues of $H_I$ at each step, which can be exponentially long on its own for the most complex $H_I$, but is not required during QA. To exploit this resource, one should create a small processor, with only ~25 high-quality qubits, but with $H_I$ that depends on ~$2^{25}$ different coupling parameters.

## Discussion

Finding the ground state of an arbitrary Ising spin Hamiltonian is generally an exponentially hard computational problem. Even harder, it seems, is to study dynamics with a time-dependent quantum Hamiltonian that implements quantum annealing computation in the nonadiabatic regime. Nevertheless, we showed that a fully solvable model for the most general case of Ising spin interactions exists.

In other branches of physics, integrable many-body models have been very influential—often not for a particular experimental application but for the opportunity to understand the behavior of complex matter in the regimes unreachable to numerical simulations. Similarly, our exact solution produces an insight into both spin-glass physics and quantum computing from an original perspective. Thus, we used it to set new limits on the computation precision and proved the better relaxation scaling of the residual energy for quantum over classical annealing computations.

Numerically, we found considerable evidence to our conjecture that in the limit of the maximal complexity of the computational problem our solvable QA protocol outperforms other protocols for arbitrary QA rate at identical conditions for the time of computation. Given also the "no-free-lunch" property of algorithms, this leads to a new conjecture that more structured computational problems can be solved by certain QA protocols faster than in our solvable model. We provided the arguments in support of this conjecture too. Hence, our analytical solution can serve as a reference for the performance that can be achievable in the nonadiabatic regime for arbitrary $H_I$.

A currently discussed technical question, besides improving quantum coherence, is how to redesign the inter-qubit connections and the annealing protocol in order to improve heuristic QA[41]. It is often stated that the performance can improve if one-to-many qubit couplings are implemented in the Ising Hamiltonian, and if the annealing protocol has a simpler spectrum in order to make it less biased and thus reduce the effects of resonances that are specific to $H_M$. Our results show that such approaches may not lead to a boost in performance. In fact, the solvability of our model follows from a high symmetry that makes the solvable protocol maximally unbiased. We showed that this provides the advantage, over other protocols, only for the tasks with the maximal complexity but not for more structured Ising spin Hamiltonians. Hence, by adding one-to-many qubit connections and preparing less biased QA protocols, we may only bring the complexity of the QA computations closer to the domain of our model's superiority.

Our findings suggest that the quantum annealing superiority, for a specific problem, over all classical algorithms should be searched either in small size processors but with combinatorially complex interactions in $H_I$ or among relatively simple-structured $H_I$, with a polynomial number of parameters but a transverse part $g(t)H_M$ that is tailor-made for this specific computational task. It is thus important to understand how the QA performance depends on the correlations between $H_I$ and $H_M$, and on the prepared correlations in the initial state for quantum annealing.

## Methods

**Solution for QA model with arbitrary target Hamiltonian**. The annealing problem is sometimes formulated so that the target Hamiltonian, $H_0$, is different from

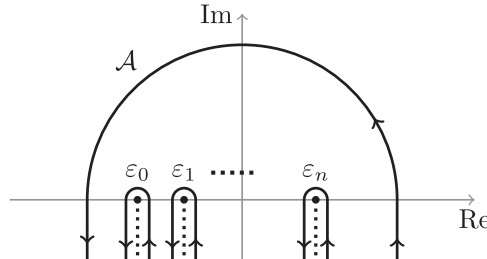

**Fig. 5 Contour $\mathcal{A}$ defined in (40).** This contour integral can be computed using the integral over the contours enclosing the branching cuts.

the Ising Hamiltonian:

$$H(t) = H_0 + g(t)H_M. \tag{37}$$

Let us show that our protocol with $g(t) = g/t$ and $H_M$ given by (7) is still solvable in the sense that we can write the probabilities of the final eigenstates of $H_0$ in terms of the parameters of $H_0$.

Suppose that $U$ is the unitary operator that diagonalizes $H_0$, i.e.,

$$H_I = UH_0U^\dagger$$

is a diagonal matrix. The latter means that it can be written in the Ising form (2), and we can define the basis states $|n\rangle$ where $n$ is the index of the excitation, as in the main text. Let us define the state

$$|\psi'\rangle = U|\psi_0\rangle, \quad |\psi_0\rangle \equiv |\rightarrow, \rightarrow, \dots, \rightarrow\rangle.$$

In the basis $|n\rangle$, the entire Hamiltonian has the form

$$H(t) = H_I - \frac{g}{t}|\psi'\rangle\langle\psi'|. \tag{38}$$

It is now almost the same as in the problem considered in the main text but the state $|\psi'\rangle$ is dependent on the matrix $U$. Hence, the matrix elements of the mixing part are given by

$$(gH_M)_{nm} = g_n g_m^*,$$

where

$$g_n = \sqrt{g}\langle n|U|\psi_0\rangle. \tag{39}$$

Thus, unlike the model in the main text, the mixing Hamiltonian $gH_M$ depends on $\mathcal{N}$ generally different parameters that depend on the eigenstates of $H_0$ via the matrix elements of $U$.

Nevertheless, the most general form of the model that was solved in Appendix B of ref. [12] includes this particular case. Thus, if we define the probabilities

$$p_n \equiv e^{-2\pi|g_n|^2}$$

then equation (12) for the excitation probabilities (see also equation (B13) in ref. [12]) is extended to

$$P_n = \frac{(1 - p_n)\prod_{k<n}p_k}{1 - \prod_{n=0}^{\mathcal{N}}p_n}. $$

Returning to the original problem (5) in the main text, it follows from (39) that knowledge of a unitary transformation $UH_IU^\dagger$, such that its action increases the overlap of the ground state with the state $|\psi_0\rangle$, can be used to increase the probability to find the ground state.

**Solution of the model**. Following steps from Appendix B in ref. [12], we perform Laplace transformation

$$a_n(t) = \int_{\mathcal{A}} e^{-st} b_n(s)\,ds, \tag{40}$$

where $\mathcal{A}$ is a contour in the complex plane such that the integrand vanishes when $\mathcal{A}$ originates and escapes to infinity (Fig. 5). Substituting (40) into (11), we find a first-order differential equation with a simple solution for $b_n(s)$, which we substitute to (40) to find

$$a_n(t) = c\int_{\mathcal{A}}\frac{e^{-st}}{-s + \varepsilon_n}\prod_{n=0}^{\mathcal{N}-1}(-s + \varepsilon_n)^{ig/\mathcal{N}}\,ds, \tag{41}$$

where $c$ is a normalization constant that is fixed by the initial conditions. Following[12], as $t \to \infty$ this integral is evaluated using the saddle point method and suitable deformation of $\mathcal{A}$ into the paths that go around the branch cuts in Fig. 5. This results in the analytical expression for $a_n(t \to \infty)$ in terms of the Gamma

| Table 1 Quantum annealing protocols with different interaction $H_M$ and schedule $g(t)$, but of the same annealing time. | | |
|---|---|---|
| **Quench Protocol** | **Schedule** | **Control Hamiltonian** |
| Protocol 1 | $-g/t$ | $H_M = \lvert\psi_0\rangle\langle\psi_0\rvert$ |
| Protocol 2 | $g/(at), a = N$ | $H_M^0 = -\sum_{k=1}^{N} \sigma_x^k$ |
| Protocol 3 | $-g/(at^2), a = \Delta E_I/g$ | $H_M = \lvert\psi_0\rangle\langle\psi_0\rvert$ |

$\Delta E_I$ refers to the bandwidth of the Hamiltonian $H_I$. See Methods for analysis of the parameters.

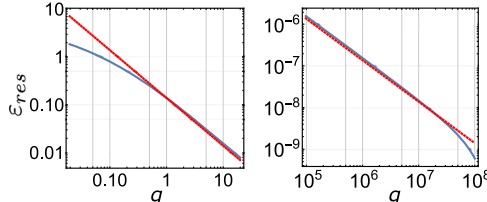

**Fig. 7 Residual energy for the exactly solvable exponential spectral density.** Blue solid line: residual energy in equation (50) at various scales. The parameters are fixed at $a = 1$ and $\mathcal{N} = 10^9$. To guide the eye, ~$1/g$ is plotted in red dotted lines.

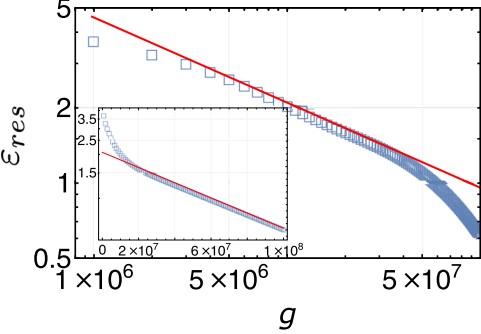

**Fig. 6 Residual energy of the Gaussian and exponential model.** Simulation of the residual energy for Gaussian (triangle) and exponential (square) spectral density, defined in equations (45, 46). Red and black solid lines are $\sim 1/\sqrt{g}$ and ~$1/g$, respectively. The total number of states is fixed at $\mathcal{N} = 10^9$.

**Fig. 8 Residual energy of the independent spin model.** Simulation of the residual energy for the independent spin model, plotted in log–log scale. The red solid line is fitted to $1/g^{0.34}$. The total number of spins is 30, which corresponds to $\mathcal{N} \approx 10^9$. Inset is the same plot in semi-log scale, which shows an exponential fit of the adiabatic tail (red solid line).

function of the parameters. The excitation probability is then obtained from $P_n = \lvert a_n(t \to \infty)\rvert^2$, and using the properties of the Gamma function.

**Setting parameters of protocols to compare their performance.** First, we note that $H$ with $H_M$ in (7) and $H_M^0$ in (17) have the same ground states both as $t \to 0_+$ and $t \to \infty$. For both of them, the maximum density of states is at zero energy. Hence, for $H_M$, $\Delta E_t = g(t)$, and for $H_M^0$, $\Delta E_t^0 = Ng(t)$, where $N$ is the number of spins. If, for the analytically solvable protocol with $H_M$, we choose the time-dependent form $g(t) = g/t$ and fix the quench parameter $g$, then the annealing time is given by $g/\tau_a = \Delta E_I$, or

$$\tau_a = g\tau_I, \tag{42}$$

where $\tau_I = 1/\Delta E_I$. This also gives the meaning to the parameter $g$, that is, the ratio of the annealing time and the characteristic time of the dephasing by $H_I$. For the transverse field protocol (7) with $H_M^0$, the same annealing time $\tau_a$ in [(42)] is achieved if we set

$$g_0(t) = \frac{g}{Nt}. \tag{43}$$

Similar arguments for $g_0(t) \sim 1/t^2$ lead to $g(t) = -g/(at^2)$, where $a = \Delta E_I/g$, as listed in Table 1.

**Scaling of the residual energy.** In the main text, we have shown that for a uniform spectral density, $\rho(E) = $ constant, the residual energy (25) scaling is a power-law in the annealing time $\tau_a$ (or in the parameter $g$).

The power-law scaling of the residual energy can be generalized to any power-law dependence of $\varepsilon_n$, by readily evaluating the average over the probability distribution (12). Namely, for $\varepsilon_n \propto n^\alpha$, and in the limit of $\mathcal{N} \gg 1$, it can be shown that $\varepsilon_{res} \sim 1/g^\alpha$.

For a generic spectral density $\rho(\varepsilon)$, the energy level index $n$ can be written as a function of the energy,

$$n(\varepsilon) = \int_{\varepsilon_0}^{\varepsilon} \rho(x)dx. \tag{44}$$

Without loss of generality, let us assume that the ground state has zero energy, $\varepsilon_0 = 0$. The residual energy can be evaluated as

$$\varepsilon_{res} \equiv \sum_n P_n \varepsilon_n = \int d\varepsilon P_{n(\varepsilon)} \varepsilon \rho(\varepsilon),$$

where $P_n$ is the probability distribution (12). The behavior of the residual energy is determined by the shape of the spectral density. However, we argue that its power-law scaling is generally expected.

Consider the Gaussian and exponential spectral densities. Both of the spectra are restricted to the energy range [0, 2] and are centered at $\varepsilon = 1$. The Gaussian spectral density we simulated is

$$\rho(\varepsilon) = Ae^{-10(\varepsilon-1)^2}, \quad 0 < \varepsilon < 2, \tag{45}$$

and the exponential spectrum is

$$\rho(\varepsilon) = \begin{cases} B(e^{10\varepsilon} - 1), & 0 < \varepsilon < 1, \\ B(e^{10(2-\varepsilon)} - 1), & 1 < \varepsilon < 2, \end{cases} \tag{46}$$

where $A$ and $B$ are normalization factors. At large annealing time, when the system approaches the ground state, we can expand the spectral density to the leading order of the energy. Note that the exponential spectrum modeled above vanishes at the ground state, hence, with (44), $n(\varepsilon) \propto \varepsilon^2$. Using the result for power-law $\varepsilon_n$ aforementioned, we expect a scaling of the residual energy $\varepsilon_{res} \sim 1/\sqrt{g}$. The Gaussian spectrum studied above has a finite value at the ground state cutoff. This constant value can dominate the sub-leading terms when the total number of states is sufficiently large. In this case, we get $\varepsilon_n \propto n$ and consequently $\varepsilon_{res} \sim 1/g$. These two types of scaling behavior are verified in Fig. 6.

We now consider an analytically solvable model case. Suppose the spectral density near the ground state is given by an exponential function

$$\rho(\varepsilon) = ae^\varepsilon, \tag{47}$$

with a finite but small density at the ground state, $\rho(0) = a$. The number of states below energy $\varepsilon$ is

$$n(\varepsilon) = \mathcal{N}a \int_0^\varepsilon dx\, e^x = \mathcal{N}a(e^\varepsilon - 1), \tag{48}$$

where $\mathcal{N}$ is the total number of states. Therefore,

$$\varepsilon_n = \log(n + \mathcal{N}a) - \log(\mathcal{N}a). \tag{49}$$

The average of this energy over the distribution $P_n$(12) can be computed exactly. We note that the exponential density of the state is only valid at small energies since it diverges when $\varepsilon$ becomes large. Hence, to get a physically sensible result representing the correct low-energy behavior, the total number of states $\mathcal{N}$ must be

sufficiently large, so the decay of $P_n$ at large $n$ compensates for the nonphysical growth of the exponential spectral density. With this, we get

$$
\begin{aligned}
\varepsilon_{res} &= \sum_0^\infty \varepsilon_n P_n \\
&= -\frac{1-p}{1-p^{\mathcal{N}}} \Phi^{(0,1,0)}(p, 0, \mathcal{N}a) - \log(\mathcal{N}a),
\end{aligned}
\tag{50}
$$

where $p = e^{-2\pi g/\mathcal{N}} \approx 1 - 2\pi g/\mathcal{N}$, $\Phi(x, y, z)$ is the Lerch transcendent function, and $f^{(0,1,0)}$ is the derivative of $f$ with respect to its second argument. This function at large $\mathcal{N}$ simplifies to a power-law scaling $\sim 1/g$ (see Fig. 7).

As the last example, consider a model of many non-interacting spin 1/2's. Each spin has eigenenergies ±1. The number of states for a fixed number of spin excitations is given by the binomial distribution. This allows us to compute the energy levels exactly. This model has a global spectral density well described by a Gaussian function. Figure 8 shows a finite size simulation of the residual energy, which scales as a power-law $1/g^\alpha$, with the exponent fitted to $\alpha \approx 0.34$. Note that $\alpha = 1/3$ is expected for a Gaussian spectrum, whose density of states vanishes at the ground state, because it expands to the leading order as $\sim \varepsilon^2$, which results in $\varepsilon_n \propto n^{1/3}$. Our simulation fits into this picture very well. The residual energy eventually switches to an exponential decay near the truly adiabatic regime (as shown in the inset). This is expected because the energy relaxation is dominated then by only a few states that all decay with time exponentially.

## Data availability

The data that support the findings of this study are available from the corresponding author on reasonable request.

## Code availability

The code used to generate the data in this study is available from the corresponding author upon reasonable request.

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

## Acknowledgements

This work was carried out under the support of the U.S. Department of Energy, Office of Science, Basic Energy Sciences, Materials Sciences, and Engineering Division, Condensed Matter Theory Program. B.Y. also acknowledges partial support from the Center for Nonlinear Studies.

## Author contributions

N.S. proposed the project. B.Y. and N.S. conducted the research and wrote the paper.

## Competing interests

The authors declare no competing interests.
