## [Peer review file · Nature Communications]

REVIEWER COMMENTS

Reviewer #1 (Remarks to the Author):

Quantum annealing is among the paradigmatic approaches of quantum computing. While it is a highly popular and broadly studied approach, it remains to a large extent heuristic. It is, in particular, true in applications to optimization problems, where a promise of advantage (or the lack of it), when compared to classical approaches, remains elusive.

The submitted contribution makes an important step towards better understanding promises of quantum annealing. It provides an analytical solution (probabilities of any classical configurations in the quantum state at the end of the quench) for a particular driver Hamiltonian and a specific quench protocol, that is valid for any classical problem Hamiltonian. The driver Hamiltonian is symmetric and non-local (including all-qubits interactions), but its high symmetry indeed allows for analytical insight. The authors also show that it has a highly desired feature of being unbiased, in the sense that all degenerate target states have the same probability. They also argue about the optimality of their strategy in some cases, which provides a new reference point for widely used heuristic approaches.

I believe that this is an important contribution that should enjoy high recognition and open new directions in this field of quantum annealing. As such, I would like to recommend working towards its publication in Nature Communications. I, however, believe some sharpening of the presentation is necessary, please see the points below:

1) The authors make a comparison of residual energy in their protocol [claimed to be inversely proportional to annealing time in Eq. (25), see also Eq. (26)] with a classical behavior that is vanishing logarithmically as in Eq. (3).

I'm not sure if this is an "apple with apple" comparison.

Quite often, an additional requirement/assumption put on (quantum) annealing is that the problem Hamiltonian should be extensive in the number of qubits -- and I'm wondering if there are no such assumptions behind Eq. (3)? How would the protocol described in the manuscript behave for extensive problem Hamiltonian? (e.g., with the expected scaling of ΔE_I with N for such H_I)

On the other hand, the authors focus primarily on "the most complex" problem Hamiltonian with exponentially many terms in the Ising formulation. The obtained scaling in Eq. (35) hinges on the behavior of the density of states possible for such a problem Hamiltonian. I'm wondering how the classical annealing would behave under such an assumption -- is Eq. (3) still valid in that case?

I believe that the authors should take special care to be clear about such assumptions throughout the paper.

2) Below Eq. (8), they cite [10] as a place where Hamiltonian in Eq. (8) also appeared. If I recall correctly, it was also considered in [13], though only concerning sufficient time for ground state preparation.

3) Below Eq. (22), the authors state that an exponentially large number of terms in H_I would add an extra exponential overhead on Monte-Carlo approaches. But wouldn't it also put a similar overhead on Quantum annealing? -- somehow, all that exponentially many terms would have to be stored and imposed also for quantum protocol. Is it a fair comparison?

4) At the beginning of Sec 4.D.2, the authors talk about "small connectivity". Please be more specific, e.g., is the Edwards-Anderson model of small connectivity in that context. How about the Sherrington-Kirkpatrick model? Also, please add a citation.

In the inset of Fig 4. we see a nice power-law, including for Sherrington-Kirkpatrick Hamiltonian. I'm wondering what would happen for larger values in g . Would protocol no. 2 transition to other behavior for significantly larger values of g ?

In the last paragraph of that subsection, the authors talk about "stiffest QA" (also in a couple of other places). It is jargon -- please be more specific.

5) There is a typo with the missing reference "[?]" below Eq. (20) and in a few subsequent places.

Reviewer #2 (Remarks to the Author):

In this paper, the authors use an analytical solution, published in a number of previous publications, to argue that scaling of the residual energy with computation time in quantum annealing (QA) is polynomial, instead of logarithmic, e.g., Eq. (3), expected for classical annealing. If true, this is a significant result because it shows that QA is exponentially faster than its classical counterpart in reducing the residual energy. Unfortunately, I believe this result is obtained based on assumptions that may not be justified, as I will discuss below.

First of all, I believe Eq. (3) is obtained under the constraint that reaching the ground state at the end of computation is guaranteed. In practice, however, if the goal of computation is just to reduce the residual energy much shorter computation times may be sufficient.

Second, although the analytical solution described in Sec. II for the specific initial Hamiltonian of Eq. (8) is impressive, the final result summarized in the equation above (21) is what you expect for an unstructured quantum search algorithm without slowing down at the minimum gap location (locally adiabatic evolution). Generalization of this result to the more physically motivated Hamiltonian of Eq. (18) is rather handwavy but may still be okay, meaning that Eq. (23) may remain valid.

Finally, the main result of this paper that is summarized in Eq. (25) and (26) is obtained under the assumption that the energy levels of the Ising Hamiltonian are linearly spaced by an energy distance $\Delta = \Delta E_1/N$, as discussed above Eq. (25). This assumption plays the key role in the conclusion that the residual energy is reduced polynomially instead of logarithmically with time. For most Hamiltonians, however, the density of states is expected to be maximally packed near the center of the spectrum with an exponential decrease as the ground state is approached. This can be shown easily for simple Hamiltonians such as Eq. (35), but may also be verified for more complex Hamiltonians. As a result, the residual energy will have logarithmic dependence on $\langle n \rangle$, instead of linear as assumed. Therefore, the dependence of the residual energy on the computation time becomes logarithmic, similar to the classical case.

I therefore believe that the main conclusion of this paper is an artifact of an unjustified assumption. For this reason, I cannot accept this paper for publication.

Here are some more specific comments:

- The index t of H_t in Eq. (1) can be confused with time.
- On page 1, the typical energy spacing is stated to be $\Delta E_1/2^N$, which is not true as discussed above.
- β in Eq. (3) which is also repeated in the abstract is undefined.
- Above Eq. (9), "other eigenstates of H_t are zero" should be "other eigenvalues of H_t are zero".

- In the unnumbered equation below (11), the energy levels are assumed to be nondegenerate.

Summary of changes:

In the revised manuscript, we highlighted the main changes in blue color. Major changes include:

1. New analysis added regarding the residual energy scaling. These are summarised in blue text on Page 5 and in the new section (the last section in **Methods**, with three new figures (Figs. 6,7,8)).
2. Previous equation (3) and its discussion in the introduction are moved to the new equation (22) and the corresponding section.
3. The last subsection in the section of **Results** is shortened in order to keep the length of the article within the Nature Communications limits.
4. Title is shortened to “Analytical solution for nonadiabatic quantum annealing to arbitrary Ising spin Hamiltonian”.
5. Structural changes according to the Nature Communications formatting guidelines.

We would like to thank both of the reviewers for their valuable comments and suggestions, which have helped us substantially improved the quality of the manuscript. Below is the point-to-point responses to the reviewers:

Responses to Reviewer #1:

Reviewer: I believe that this is an important contribution that should enjoy high recognition and open new directions in this field of quantum annealing. As such, I would like to recommend working towards its publication in Nature Communications. I, however, believe some sharpening of the presentation is necessary, please see the points below:

Response: We are grateful to Reviewer 1 for very positive response. We address all Reviewer’s suggestions/questions below.

Reviewer: 1) The authors make a comparison of residual energy in their protocol [claimed to be inversely proportional to annealing time in Eq. (25), see also Eq. (26)] with a classical behavior that is vanishing logarithmically as in Eq. (3). I’m not sure if this is an ”apple with apple” comparison. Quite often, an additional requirement/assumption put on (quantum) annealing is that the problem Hamiltonian should be extensive in the number of qubits – and I’m wondering if there are no such assumptions behind Eq. (3)? How would the protocol described in the manuscript behave for extensive problem Hamiltonian? (e.g., with the expected scaling of ΔE_I with N for such H_I).

Response: The logarithmically slow relaxation in time is the feature of all systems that are called glasses. So, as a phenomenological consequence, a similar logarithmic dependence on the annealing time in Eq. 3 (now Eq. 22) has been expected to be true for all glass systems - not only extensive systems.

Our solution addresses the systems of both type equally well. Hence the scaling laws that we found are equally applicable to both strongly connected and lattice-like spin systems.

Reviewer: On the other hand, the authors focus primarily on "the most complex" problem Hamiltonian with exponentially many terms in the Ising formulation. The obtained scaling in Eq. (35) hinges on the behavior of the density of states possible for such a problem Hamiltonian. I'm wondering how the classical annealing would behave under such an assumption – is Eq. (3) still valid in that case? I believe that the authors should take special care to be clear about such assumptions thought the paper.

Response: We would like to respectfully disagree with Reviewer. The scaling in Eq. 26 (new Eq. 24) is found in our solution for all possible Ising Hamiltonians. Moreover, for energy relaxation it does not depend on the system size. So, these equations are equally applicable to lattice-like spin systems as to random spin clusters. Until the following subsection, our discussion was completely general, without assuming the limit of maximal complexity.

However, we agree with Reviewer that the generality of Eqs. 25 for the residual energy (new Eq. 26) has to be discussed additionally. The resubmitted manuscript now explains that the typical systems that have spectra with a decreasing density of states when approaching the ground energy, do not show a logarithmic residual energy dependence on the QA time but they typically show a power-law relaxation, with exponential tail at the transition to the truly adiabatic regime.

Reviewer: 3) Below Eq. (22), the authors state that an exponentially large number of terms in H_I would add an extra exponential overhead on Monte-Carlo approaches. But wouldn't it also put a similar overhead on Quantum annealing? – somehow, all that exponentially many terms would have to be stored and imposed also for quantum protocol. Is it a fair comparison?

Response: We agree with Reviewer that this was a point that needed more explanation. Hence, we added extra discussion to the main text.

Rough estimates are as follows. If we have exponentially many, $\sim 2^N$, terms in the Hamiltonian, then to find only one eigenvalue we need $\sim 2^N$ time. So the time to sort the array of eigenvalues takes 2^{2N} steps.

On the other hand, for QA, it takes about 2^N steps to set couplings between the qubits, and then the annealing time $\sim 2^N$ to perform QA. Hence, the net time still scales as $\sim 2^N$, i.e., QA is much faster.

Reviewer: 4) At the beginning of Sec 4.D.2, the authors talk about "small connectivity". Please be more specific, e.g., is the Edwards-Anderson model of small connectivity in that context. How about the Sherrington-Kirkpatrick model? Also, please add a citation.

Response: Here "small connectivity" refers to what is sometimes called generalized SK-model. — We agree with Reviewer that the two aforementioned models are worth mentioning. In the revised manuscript we have clarified this point and added new references [20,21,32]

Reviewer: In the inset of Fig 4. we see a nice power-law, including for Sherrington-Kirkpatrick Hamiltonian. I'm wondering what would happen for larger values in g . Would protocol no. 2 transition to other behavior for significantly larger values of g ?

Response: We also would be interested. However, such questions are hard to answer because explicitly time-dependent simulations are very time-expensive even for relatively low (12 in our case) spins. We did statistical averaging for this figure using a LANL's supercomputer, and computation time is proportional to g . So, reaching a significantly larger g values without computation errors, unfortunately, is beyond our numerical capabilities. This is why the models like our's

are needed - to make an insight into the numerically inaccessible regimes.

On the other hand, in the revised manuscript we added a new section (the last section in **Methods**) which includes discussion of the scaling of the residual energy. A few more model studies are included which do not require brute force computations. This allows us to reach significantly large values of g . In these cases power-law scaling always holds.

Reviewer: In the last paragraph of that subsection, the authors talk about "stiffest QA" (also in a couple of other places). It is jargon – please be more specific.

Response: We agree with Reviewer, in the new version we define the model with all nonzero random parameters as the "maximal complexity" case, and we avoided using "stiff" terminology.

Reviewer: 5) There is a typo with the missing reference "[?]" below Eq. (20) and in a few subsequent places.

Response: The citation errors have been fixed.

Responses to Reviewer #2:

Reviewer: In this paper, the authors use an analytical solution, published in a number of previous publications,

Response: Please let us clarify here. We were writing not one more article on some famous model. The solution of our model has been published by one of us in appendix to a paper in 2014 as an illustration of a certain mathematical concept. Since then, it has never been reproduced elsewhere and has never been applied to physics or information science. So, we hope Reviewer can see that identifying importance of this solvable model for quantum annealing is a nontrivial step on its own. In fact, it took some steps in our article to connect the solvable model in the previously known form to the considered quantum annealing problem. Moreover, only one formula for the final state probabilities was used in our present article. All other formulas that we wrote are not found in the original solution.

to argue that scaling of the residual energy with computation time in quantum annealing (QA) is polynomial, instead of logarithmic, e.g., Eq. (3), expected for classical annealing. If true, this is a significant result because it shows that QA is exponentially faster than its classical counterpart in reducing the residual energy. Unfortunately, I believe this result is obtained based on assumptions that may not be justified, as I will discuss below.

Response: As we will detail below, we would like to respectfully disagree with some of the Reviewer's arguments. However, we agree that they are worth a discussion, so we also made proper changes in the main text to avoid similar critics from the readers.

Reviewer: First of all, I believe Eq. (3) is obtained under the constraint that reaching the ground state at the end of computation is guaranteed. In practice, however, if the goal of computation is just to reduce the residual energy much shorter computation times may be sufficient.

Response: We would agree with Reviewer in that one can derive a *bound on the annealing schedule*, $\sim 1/\log T$, where T is the classical annealing time, so that if the annealing is performed slower than this, the ground state is guaranteed to be reached. We also agree that at short time, the relaxation is not universal and can be much faster. Eq. 3 (now Eq. 22) has been always used

to describe long-time tail of the relaxation and only in spin glasses. This is the regime in our focus.

However, the logarithmic $\sim 1/\log^\beta T$ with $\beta > 1$ scaling refers to the time-dependent relaxation of the residual energy for classical annealing in a very large class of glass systems. It is the scaling that is actually used to define a transition to the glass phase. So, if we believe in glasses, we believe in this logarithmic law.

A more complex question is whether this law is expected for quantum annealing through a glass phase. We cited the literature that suggested the same logarithmic law for quantum annealing, with a different interpretation of T . In numerous prior numerical studies, such a logarithmic energy relaxation during finite time quantum annealing through a glass phase has always been confirmed. So, now it is generally accepted, and has been even supported by non-rigorous but analytical arguments.

It is fair to compare with our $1/T$ energy relaxation as our solution applies to arbitrary Ising spin glasses in the same context as the previously conjectured logarithmic law.

We cited only a few papers in the original version about the logarithmic law. This could be not enough. In the revision we have extended the bibliography to review this topic.

Reviewer: Second, although the analytical solution described in Sec. II for the specific initial Hamiltonian of Eq. (8) is impressive, the final result summarized in the equation above (21) is what you expect for an unstructured quantum search algorithm without slowing down at the minimum gap location (locally adiabatic evolution).

Response: Although our time estimates for a general H_I may look not impressive, first, we note that it is better than the result expected from classical annealing of spin glasses. This proves that quantum annealing is free of certain drawbacks of classical annealing, which often performs worse than a brute-force random search.

Second, we show that this result does not apply to the case with ground state degeneracy, for which even exponential speedup can be achieved to find a specific ground state.

So, our findings are actually consistent with common belief that quantum algorithms cannot solve “typical” computational problems but can give strong boost in rare cases, e.g., by using collective effects that are often provided by systems with strong energy degeneracy.

Reviewer: Generalization of this result to the more physically motivated Hamiltonian of Eq. (18) is rather handwavy but may still be okay, meaning that Eq. (23) may remain valid.

Response: We think that there is a misunderstanding here. We did compare our solvable protocol to the protocol with a decaying transverse field but we do not claim that the solvable result applies to the transverse field case.

In fact, our conclusions based on this comparison are very different. On one hand, we show that the solvable protocol outperforms the transverse field in the maximal complexity limit and in situations with the ground state degeneracy. On the other hand, we show that the transverse field clearly outperforms the solvable one on small-connectivity spin glasses.

So, our solution is not to apply to quantum annealing generally. Instead, it can be used to set certain limits on what is possible and disprove hypotheses such as the logarithmic energy relaxation in spin glasses.

Reviewer: Finally, the main result of this paper that is summarized in Eq. (25) and (26) is obtained under the assumption that the energy levels of the Ising Hamiltonian are linearly spaced

by an energy distance $\delta = \Delta E_I/N$, as discussed above Eq. (25). This assumption plays the key role in the conclusion that the residual energy is reduced polynomially instead of logarithmically with time. For most Hamiltonians, however, the density of states is expected to be maximally packed near the center of the spectrum with an exponential decrease as the ground state is approached. This can be shown easily for simple Hamiltonians such as Eq. (35), but may also be verified for more complex Hamiltonians. As a result, the residual energy will have logarithmic dependence on $\langle n \rangle$, instead of linear as assumed. Therefore, the dependence of the residual energy on the computation time becomes logarithmic, similar to the classical case.

Response:

First, we would like to respectfully disagree with Reviewer about the generality of Eq. 26 (Eq. 24 in the resubmitted version). It is for the scaling of the average number of excitations - not residual energy. Hence, this result is exact and valid for any energy dispersion.

For Eq. 25 (Eq. 26 in the new version), in the previous version we wrote that the deviations for the residual energy are expected but not of the logarithmic type. In the new version, we provide more evidence to this claim.

We agree with the referee that, near the ground state, an exponential tail of the density of states results in a logarithmic dependence of the energy E_n to the level index n , e.g., $E_n \sim \log n$. However, $\langle E_n \rangle \equiv \langle \log n \rangle \neq \log \langle n \rangle$. In fact, our exact solution for the residual energy with an exponential spectral density of the form $\rho(E) = ae^E$ near the ground state still reveals a power-law relaxation. (See the last section of **Methods**.)

The average is very sensitive to the form of the probability distribution P_n . To see this fact, consider a simple extreme example, where, beside the ground state with energy E_0 , all other states have the same energy E_1 . Hence the spectrum is sharply concentrated at the highest energy. In this case, the residual energy is

$$\epsilon_{res} = P_0 E_0 + (1 - P_0) E_1 - E_0 = (1 - P_0) \Delta E,$$

where P_0 is given by Eq. (13) in the manuscript, hence,

$$\epsilon_{res} \sim e^{-\frac{2\pi}{N} \frac{\tau_a}{\tau_I}}, \quad (1)$$

which means an exponential decay of the residual energy with the annealing time τ_a . Therefore, even though the exact form of the spectral density near the ground state becomes important, special care must be given when taking the average for the residual energy.

We note that the power-law scaling of the residual energy is valid for any power-law energy dispersion. Namely, for $E_n \sim n^\alpha$, we have $\epsilon_{res} \sim 1/g^\alpha$, where $\alpha = 1$ corresponds to the special case of uniform spectral density. This allows us to study the residual energy with general forms of the density of states close to the ground state. In the revised manuscript, we added a new subsection in the section of **Methods** with more comprehensive discussions. We have shown there that a power-law relaxation of the residual energy is generally expected. This is verified by our simulations including for both Gaussian and exponential density of states.

Reviewer: The index t of H_t in Eq. (1) can be confused with time.

Response: We have changed the notation to H_M with M referring to “mixing”.

Reviewer: On page 1, the typical energy spacing is stated to be $\Delta E_I/2^N$, which is not true as discussed above.

Response: Strictly speaking, by definition, ΔE is the interval that contains almost all energy states. Average spacing usually coincides with typical. However, we agree that this estimate does not have to apply to the minimal gap from the ground state that may encounter during quantum annealing. The latter for spin glasses (about which we were talking) is still typically exponentially small, although with a different from $1/2^N$ suppression factor. We rewrote our discussion in introduction to be more precise.

We also note that there are systems with really large gaps to the ground state - without exponential suppression - but they are usually not considered as true spin glasses.

Reviewer: β in Eq. (3) which is also repeated in the abstract is undefined.

Response: β is a system dependent scaling factor. In the revision we have clarified this.

Reviewer: Above Eq. (9), "other eigenstates of H_t are zero" should be "other eigenvalues of H_t are zero".

Response: We are grateful to Reviewer for pointing to this typo. It is now fixed.

Reviewer: In the unnumbered equation below (11), the energy levels are assumed to be nondegenerate.

Response: We treat the case with degeneracy separately. However, for random spin glasses the lack of degeneracy is expected. So, we just mentioned the issue of degeneracy at the place where this equation appears, and left discussion of the degeneracy case to a later section.

REVIEWERS' COMMENTS

Reviewer #1 (Remarks to the Author):

I believe that the presentation in the revised version of the manuscript is much sharper. All the issues I raised in my previous report have been properly addressed.

An analytical solution that allows assessing the performance of quantum annealing is extremely valuable. Additionally, the manuscript provides a general critical discussion contrasting the pros and cons of the soluble protocol compared with commonly used heuristic approaches. The paper has the potential to guide the effort to develop quantum annealers. As such, I would like to reiterate my support for the publication of this article in Nature Communications.

A minor issue that captured my attention at the end of the "Avoiding the bound" section. The authors discuss an additional example with Hamiltonian in Eq. (36) [it is not crucial to the overall discussion in the paper, and, if necessary, could be dropped]. The authors compare it to the classical sorting with complexity $O(N * \log(N))$. However, if I'm not mistaken, a partitioning of the list of ϵ_k into the larger $N/2$ and smallest $N/2$ should be sufficient (without sorting). In that case, this operation requires $O(N)$ time.

Also, in the method section, the authors forgot to update notation, changing H_t to H_M . There seems to be a typo in Eq. (47) where ϵ does not appear on the right-hand side.

Reviewer #2 (Remarks to the Author):

I am pleased with the revised version and authors' explanations. The added materials, especially the last section of Methods, are helpful and necessary for explaining how a power-law scaling of the residual energy can be expected for more general forms of density of states. I, however, believe this is a consequence of nonlocal nature of Hamiltonian (7). For any local Hamiltonian, such as (17), polynomial decay of the residual energy is only expected for second order quantum phase transitions. A first order quantum phase transition, which is expected for most realistic problems especially spin glasses, would make the computation exponentially slow. This means the residual energy would decay logarithmically with time. This point needs to be stressed to avoid misunderstanding. Nevertheless, I now believe these

results, despite having limited applicability to realistic situations, are interesting and important enough to deserve publication.

Reviewer 1: A minor issue that captured my attention at the end of the "Avoiding the bound" section. The authors discuss an additional example with Hamiltonian in Eq. (36) [it is not crucial to the overall discussion in the paper, and, if necessary, could be dropped].

Response: Reviewer 1 proposed an optional change of removing the example discussed in the section of "Avoiding the bound".

We have carefully thought about this suggestion. We agree with reviewer that this example is somewhat optional. However, it is needed, simultaneously, to connect our work to other literature and it helps the reader to avoid certain misunderstandings about the generality of our main model's conclusions. Besides, it does not take too much space. Therefore, we decided to keep this example withing the section "Avoiding the bound".

Reviewer 1: The authors compare it to the classical sorting with complexity $O(N * \log(N))$. However, if I'm not mistaken, a partitioning of the list of ϵ_k into the larger $N/2$ and smallest $N/2$ should be sufficient (without sorting). In that case, this operation requires $O(N)$ time.

Response: We thank the reviewer for this comment. In the revised manuscript we have added clarifications and dropped the " $\log(N)$ " as suggested.

Reviewer 1: Also, in the method section, the authors forgot to update notation, changing H_t to H_M . There seems to be a typo in Eq. (47) where ϵ does not appear on the right-hand side.

Response: We have updated the notations and fixed the typos.

Reviewer 2: I, however, believe this is a consequence of nonlocal nature of Hamiltonian (7). For any local Hamiltonian, such as (17), polynomial decay of the residual energy is only expected for second order quantum phase transitions. A first order quantum phase transition, which is expected for most realistic problems especially spin glasses, would make the computation exponentially slow. This means the residual energy would decay logarithmically with time. This point needs to be stressed to avoid misunderstanding.

Response: We agree with the reviewer's intuition but at this stage it is hard to add a proper discussion without making our presentation too unsupported or lengthy. We addressed it by adding a sentence: "We leave the question open: whether this behavior is a consequence of the non-local nature of the mixing Hamiltonian (7)." We also note, that our presentation does not contradict to the reviewer's statement.

Summary of changes.

- (i) We corrected several typos, including those that were noticed by reviewer 1;
- (ii) we rewrote Abstract to fulfil Nat. Comm. requirements.
- (iii) We added the last sentence in section "Scaling for the average excitation number" in order to address suggestion of reviewer 2.
- (vi) We also removed last sentence from Discussion, as it was not important but took much space before.